# Effects of Social Facilitation and Introduction Methods for Cattle on Virtual Fence Adaptation

**DOI:** 10.3390/ani14101456

**Published:** 2024-05-14

**Authors:** Pernille Arent Simonsen, Niels Søborg Husted, Magnus Clausen, Amalie-Maria Spens, Rasmus Majland Dyrholm, Ida Fabricius Thaysen, Magnus Fjord Aaser, Søren Krabbe Staahltoft, Dan Bruhn, Aage Kristian Olsen Alstrup, Christian Sonne, Cino Pertoldi

**Affiliations:** 1Department of Chemistry and Bioscience, Aalborg University, Fredrik Bajers Vej 7H, 9220 Aalborg Øst, Denmark; nhuste21@student.aau.dk (N.S.H.); mclaus20@student.aau.dk (M.C.); aich21@student.aau.dk (A.-M.S.); rdyrho21@student.aau.dk (R.M.D.); ithays21@student.aau.dk (I.F.T.); maaser20@student.aau.dk (M.F.A.); sstaah20@student.aau.dk (S.K.S.); db@bio.aau.dk (D.B.); cp@bio.aau.dk (C.P.); 2Skagen Bird Observatory, Fyrvej 36, 9990 Skagen, Denmark; 3Department of Nuclear Medicine, Aarhus University Hospital, Palle Juul-Jensens Boulevard 99, 8200 Aarhus, Denmark; aagealst@rm.dk; 4Department of Clinical Medicine, Aarhus University, Palle Juul-Jensens Boulevard 99, 8200 Aarhus, Denmark; 5Department of Ecoscience, Aarhus University, Frederiksborgvej 399, 4000 Roskilde, Denmark; cs@ecos.au.dk; 6Aalborg Zoo, Mølleparkvej 63, 9000 Aalborg, Denmark

**Keywords:** animals, virtual fencing, Nofence©, Angus cattle, social facilitation

## Abstract

**Simple Summary:**

Virtual fencing is an alternative to the ubiquitously used physical fences. It uses GPS to determine the location of livestock relative to a virtual line. The system deters livestock from leaving the enclosure by using an auditory warning followed by an electric impulse during approach and crossing of the fence line. This paper assessed the social effects on learning between cattle with and without prior experience with virtual fencing through three case studies: one with a gradual and two with an instant introduction method. These were applied to assess if improvements in associative learning could be made. Due to the non-standardised experiments, it was not possible to analyse the effect across the case studies and therefore to conclude whether the number of experienced cattle or the different introduction methods influenced the inexperienced cattle.

**Abstract:**

Agricultural industries rely on physical fences to manage livestock. However, these present practical, financial, and ecological challenges, which may be solved using virtual fencing. This study aimed to identify how experienced cattle through social facilitation and the introduction method influence inexperienced cattle. Based on three stocks held in Fanø, Denmark, containing 12, 17 and 13 Angus (*Bos taurus*), we examined the virtual fence learning in three case studies using one gradual introduction with zero experienced cattle (A) and two different instant introductions with one (B) and ten (C) experienced cattle. Gradual introduction had the virtual fence moved 20 m every other day for eleven days, and in the two instant introductions, the physical fence was removed in one day. Warnings and impulses were recorded during an 11-day learning period and a 26-day post-learning period, using the impulses per warning to quantify if the cattle adapted. Case studies A and B showed a significant reduction in the warnings and impulses, but only A showed a significant reduction in the impulses per warning when comparing the learning period to the post-learning period. Due to the non-standardised experiments, it was not possible to conclude if the number of experienced cattle or the introduction method had an effect on the results.

## 1. Introduction

Cattle production is a major contributor to European agriculture [1]. Managing cattle requires fencing to contain the cattle in a specific area; however, physical fences present several issues for wildlife and ecosystems. Physical barriers can cause direct effects, such as collision and entanglement, leading to an increased mortality risk [2]. Further indirect effects include habitat loss, mother–offspring separation, and fragmentation, which reduce the carrying capacity, thereby negatively affecting biodiversity [2,3,4].

Fences present both practical and financial challenges, including the monetary cost associated with installing, repairing, and relocating fences [5]. The practical aspect of not being able to build fences on certain types of terrain is also problematic [5]. These factors must be considered when establishing a pasture and managing a herd. To circumvent these challenges, virtual fences can be implemented while potentially improving welfare for cattle [6].

Nofence© (Molde, Norway) (www.nofence.no, accessed on 12 March 2024) is a virtual fence system for grazers, such as cattle and goats. It makes dynamic grazing possible, and it eliminates the need for a physical fence [7]. The concept of a virtual fence works by emitting an auditory warning from a collar when an animal approaches the virtual fence. If the animal continues, it will receive an electrical impulse. Nofence© aims to prevent the electrical impulse through a change in movement when the animal receives an auditory warning [7].

Previous studies on implementing virtual fencing suggest no substantial impact on the behaviour and welfare of several species of livestock [6,8,9,10]. Cattle can adapt to a virtual fence, as they show the ability to respond to the auditory warning and thereby receive fewer electrical impulses over time [11]. Associative learning is an essential tool in learning where cattle associate the two conditions: warning and impulse [12]. A previous study illustrates cattle’s capability to associate an auditory warning and an impulse after a week’s training, illustrating associative behaviour [12]. Studies also document variability between individuals when assessing learning ability as measured by the number of warnings and electrical impulses [11]. However, as cattle are herd animals, their ability to learn cannot be determined individually since social facilitation can influence learning [13,14]. Social facilitation (also called allelomimicry or contagious behaviour) occurs when an individual copies behaviour by another individual [15].

This study examines if social facilitation influences learning and if it can be accelerated. This is achieved by studying two different kinds of introduction methods and the presence of experienced cattle (cattle that have previously grazed with Nofence©).

The following hypotheses were tested: (i) fewer warnings, impulses, and impulses per warning are expected during the post-learning period in comparison to the learning period; and (ii) fewer warnings, impulses, and impulses per warning are expected with experienced cattle present from the learning- to the post-learning period. The impulses per warning are the total number of impulses received by an individual divided by the total number of warnings received.

## 2. Materials and Methods

This study took place on the island of Fanø, located off the west coast of Denmark in the Wadden Sea area. A farmer implemented the Nofence© system on his Angus cattle (*Bos taurus*) (Figure A1, Appendix A). Due to the varying time frames of the case studies, the study period was divided into two parts: an 11-day learning period followed by a 26-day post-learning period. During these timeframes, the pasture sizes remained similar across the case studies; therefore, these periods were selected to standardise the case studies. Two different locations were used for three different case studies: A, B and C. A and C were in the same location but two years apart. Management of the virtual fence lines was conducted by the owner of the cattle, and the collection of data was performed by Nofence©. Terms used in the article are listed and defined, see Table 1.

### 2.1. The Virtual Fencing System

Data were collected using collars developed by Nofence©. Each cow had a collar fitted around the neck with a unique serial number to identify the animal. The collar weighed 1446 g and consisted of a silicone strap, two chains connected to a GPS receiver and two solar panels [7]. The GPS receiver collected data, including the warnings and impulses that the cattle received when encountering the boundaries. The positional data were collected every 15 min and the activity data every 30 min. Activity was measured by a step counter. The position of the animal was collected whenever a warning or an impulse was given, or when the fence status changed. All the data were available on an app (Nofence), which also made it possible to move the virtual fence [7].

When the cattle approached the virtual boundary, the collars sent out an auditory warning. The auditory warning consisted of a tone that increased in pitch for 5–20 s depending on the heading and speed of the cow approaching the boundary. If the animal ignored the auditory warning and continued towards or outside the virtual boundary, the collar provided an electrical impulse of 0.2 joules at 3 kilovolts for 1 s, with a maximum of three such impulses before sending a message to the owner that the cow had escaped [7]. Three different case studies (A, B and C) of cattle’s adaptation to virtual fencing were conducted at two different locations. Two methods of introducing the cattle to the virtual fence were investigated. In the first method, called gradual introduction, one electrical fence line was replaced by the virtual fence, which was then moved incrementally (20 m, 3 times over a period of 14 days). After 14 days, the entire physical fence was replaced with a virtual fence on all sides. In the second method, an instant introduction to the virtual fence system instantly replaced the physical electrical fence. Two slightly different methods of instantaneous introduction were investigated: an electrical fence was replaced with virtual borders from one day to another or a small number of cattle from a pasture enclosed by an electrical fence were relocated to a pasture consisting of only virtual fences.

### 2.2. Experimental Protocol

#### 2.2.1. Case Study A (Gradual Introduction with Zero Experienced Cows)

Case study A was located on the east side of Fanø (named Albuen). The two types of primary habitats were dry heathland to the west and meadows to the southeast. The meadows were dominated by grasses and sedges, and the drier heathland was dominated by heather. On 28 May 2021, 12 cattle (age: 5 years) were placed in a physical electrical enclosure of 6.5 hectares. After two days, one of the four fence lines was replaced by a virtual fence (learning period started). The removed fence line was facing southwest. The border of the virtual fence was moved 20 m further south 3 times: after 6 days, 9 days and 12 days, respectively. On 10 June 2021, the three remaining physical sides of the fence were removed, leaving only virtual borders, and the area was expanded to 35 hectares (Appendix A, Figure A2) (the beginning of the post-learning period). To standardize the experiment, data collection for this and the other learning periods was kept to the first 11 days of the learning period (30 May to 8 June 2021). The following post-learning period took place for 26 days (12 June to 8 July 2021) (Appendix A, Figure A3). During these 26 days, the pasture expanded from 35 to 37.7 hectares.

#### 2.2.2. Case Study B (Instant Introduction with One Experienced Cow)

Case study B was located on the west side of Fanø (named Gåsehullerne). The area mainly consisted of grey dunes and humid dune slacks, dominated by heather and creeping willow, respectively. To conduct the experiment, 16 heifers (age < 2 years) and 1 experienced cow (age: 11 years) were placed in a physical electrical fence of 5.6 hectares on 6 May 2023. To prevent interference from the experienced cow, it was excluded from the data, but relevant data can be found in Appendix B. On 29 May 2023, the entire physical fence was removed and replaced with a virtual fence. On 30 May 2023, the area was expanded to 5.8 hectares, and on 31 May 2023 (in the morning), to 6.0 hectares. On the same afternoon, the cattle were moved to a new 5.6 hectare area. This new area was expanded to 6.2 hectares on 3 June 2023. The learning period was between 29 May and 9 June 2023, (Appendix A, Figure A4). In the following 26-day post-learning period (15 June 2023 to 11 July 2023), the area varied from 42.5 to 61.4 hectares (Appendix A, Figure A5).

#### 2.2.3. Case Study C (Instant Introduction with 10 Experienced Cattle)

Case study C was located on Albuen, sharing the same individuals as in case study A. Two cows were removed from the pasture on 1 October 2022 and 3 November 2022, respectively. A total of 1 inexperienced cow (age: 2 years) per day was added to the herd of 10 experienced cattle on 7, 9, and 11 November 2022, totalling 3 inexperienced cattle. To prevent interference from the experienced cattle, only data from the inexperienced cattle were used for data analyses, but data from experienced cattle can be found in Appendix B. The area was kept constant at 67.8 hectares throughout the experiment (Appendix A, Figure A6). The learning period was restricted to 11 days (7, 9, 11 November to 18, 20 and 22 November 2022) and the post-learning period to 26 days (18, 20, 22 November to 14, 16 and 18 December 2022).

### 2.3. Statistical Analysis

All the figures and analyses were conducted in RStudio (https://posit.co/products/open-source/rstudio/ accessed on 10 May 2024). The following tests were conducted. Boxplots were made to visualise the warnings given per day, impulses given per day and impulses given per warning. The impulses per warning is a ratio calculated by dividing the impulses by the warnings received by an individual and is a ratio indication of failed attempts to respond to a warning. The failed ratio is a number between 0 and 1 and is used as a measurement of improvement.
$$Impulses\;per\;Warning=\frac{Impulse}{Warning}$$

The data illustrated the skewness of the distributions and outliers; therefore, the median, median absolute deviation (MAD), and non-parametric tests were used. To test for differences in the medians within all the case studies, a Wilcoxon rank-sum test was carried out. Levene’s test was performed to test for the significance of the variance between impulses given per warning for each case study. A significance level of *p* < 0.05 was used to determine if the null hypothesis was rejected.

## 3. Results

After the cattle went through a learning period, the warnings per day and impulses per day were significantly lower for case studies A and B. The impulses per warning was only significantly lower in case study A.

### 3.1. Warning Frequencies

The cattle in all three case studies received fewer warnings during the post-learning period than their corresponding introduction method (Figure 1).

Significant differences in the medians were found for A and B (post-learning < learning) when the learning- and post-learning periods were compared, but not for C (Table 2).

### 3.2. Impulse Frequencies

The cattle in all three case studies received fewer impulses during the post-learning period than their corresponding learning period. The cattle in case study A had the lowest number of impulses per day in both the learning- and post-learning periods, with a median of 0.45 and 0, respectively (Figure 2).

Significance in terms of lowering the impulses per warning was found between the learning- and post-learning period for A and B (post-learning < learning), but not for C (Table 3).

### 3.3. Impulses per Warning for Introduction Methods

The impulses per warning for A in the post-learning period had a median of 0 impulses/warning, while B had a median of 0.17 impulses/warning and C had a median of 0.12 impulses/warning. The lowest impulses per warning was found in the post-learning period for A, and the highest impulses per warning was found in the learning period for C (Figure 3).

Case study A was significant between the learning period and the post-learning period. Significance between the medians was found for A, but not for B or C. However, the variance was not significant in any of the case studies (Table 4).

## 4. Discussion

### 4.1. Experienced Cattle’s Effect on Learning

It was expected that the presence of experienced cattle would enhance the adaptation of the inexperienced cattle to the virtual fence by reducing the warnings, impulses, and impulses per warning between the learning- and post-learning periods. In the case studies with zero experienced cattle (case study A) and with one experienced cow (case study B), there was a significant reduction in the warnings and impulses per day when comparing the learning period to the post-learning period. Only case study A had a significant reduction in the impulses per warning. However, as many parameters affect the results, comparing the number of experienced cattle’s effect on the impulses per warning across the three case studies is not possible.

According to Howery et al. (2000), cattle can make environmental associations when foraging, and it could be speculated that cattle can make associations with the location of fence lines [16]. Potentially, group avoidance of the fence might be occurring, and therefore, there might not be a sufficient frequency of individual stimuli taking place to learn associatively [17,18]. This suggests that the number of warnings, impulses, and impulses per warning could be reduced for reasons other than the presence of experienced cattle.

Previous studies determined that cattle exhibit social facilitation in response to virtual fences and foraging. Keshavarzi et al. (2020) showed that cattle were able to respond to virtual fence lines as a herd rather than as individuals, showing social facilitation through a leader cow [14]. Bailey et al. (2000) demonstrated that cows with previous knowledge of food rewards in a maze can function as social models and provide visual cues to other animals as to where these food rewards are located [19]. This further validates cattle’s ability to socially facilitate in response to an experienced cow.

It was not possible to show social facilitation in relation to virtual fencing in any of the case studies. This may be explained by a study by Marini et al. (2020), which suggests that controlling the movement of sheep requires at least 66% of the sheep to be fitted with a collar to discourage the non-collared sheep from crossing the virtual border [20]. One of the current study’s percentages of experienced cattle to inexperienced was not adequate, with only one experienced cow present. A percentage of 72.7% was measured with ten experienced cattle, which according to Marini et al. (2020) should be efficient for herd management [20]. The results of this study show the opposite, meaning that other factors must influence the results. It is important to be critical when comparing the behaviour of two different species. It was found in a previous study that behavioural differences in grazing and resting periods occur between sheep and cattle, but the movement patterns remain similar [21]. This creates a parameter as the behaviour between these species is not directly comparable. Furthermore, differences in intraspecific behaviour between cattle breeds, such as grazing and movement patterns, create yet another variable [22].

Stressors may contribute to behavioural changes in cattle. Regrouping has been proven to induce these changes [23]. This may explain why the current study shows no social facilitation when inexperienced cattle are introduced into an established herd, which may cause stress. Furthermore, another study indicates that cattle exhibit inter-individual differences, which may play a role in the behaviour of cattle, further influencing interactions with the virtual fence [12]. However, the results of the current study showed no significant difference in the variance of the impulses per warning, meaning that the behaviour of the herds was homogeneous.

It is important to consider specific parameters when introducing inexperienced cattle to the virtual fence. To determine the actual effect of experienced cattle, a standardised experiment has to be conducted with limited parameters and more replicates in order to make comparisons across investigations. It is also important to draw distinctions if the social facilitation is establishing an association between impulse and warning, or if the herd is simply mimicking [15].

### 4.2. The Effect of the Introduction Method

It was expected that the warnings, impulses, and impulses per warning would be reduced from the learning period to the post-learning period. In the case studies with gradual introduction (case study A) and instant introduction (case study B), there was a significant reduction in the warnings and impulses per day when comparing the learning period to the post-learning period. Only case study A had a significant reduction in the impulses per warning. However, as many parameters affect the results, comparing the introduction method based on the impulses per warning across the three case studies is not possible.

Confessore et al. (2022) used a similar introduction method to that in case study A and found a significant decrease in the impulses and warnings [24], indicating that a gradual introduction is beneficial in terms of virtual fence adaptation. The method of evaluating learning ability is seemingly as important as the introduction method. Hamidi et al. (2024) examined different evaluation methods with regard to virtual fence learning through a gradual introduction. The study examined not only the relation between impulse and warning but also the behavioural reaction of the animals and the virtual fence collars switch from learning mode to operant mode, finding that all the approaches were successful in determining virtual fence learning. Moreover, they observed two types of learning: avoidance learning and sustainable learning [18]. In this study, we did not examine which type of learning occurred, and therefore, no distinction could be made as to which type of learning is taking place. In this current study, we can only acknowledge if the cattle learn by avoidance, since no further experiment was performed to determine sustainable learning.

Another method of optimising the learning is an individual introduction rather than an introduction as a group [25]. It is difficult for a cow in a group to adapt to the system, as the stimuli received by other cattle lessen the association between auditory warning and impulse. However, this method is impractical in a commercial setting [25], as it requires separate pastures.

Even though the results of the current study could not be used to determine what method was the most favourable for the cattle, Confessore et al. (2022) found that a gradual introduction comparable to that in case study A is more beneficial [24].

### 4.3. Standardisation and Limitations

Multiple parameters appear to interfere with the results of this study. Previously mentioned parameters such as the introduction method and herd behaviour play a significant role in adaptation to the virtual fence. However, other factors, such as season variation, herd size, and change in pasture, can impact the behaviour of cattle, impeding social facilitation and interactions with a virtual fence, thereby creating further uncertainties [26,27,28].

Further studies should be conducted to determine the full effect of herd behaviour. A study could be created to analyse movement through GPS coordinates. If the cattle remain in a close herd when interacting with the virtual fence, this suggests that the cattle react as a herd rather than individually. In contrast, if the movement of the herd was generally more dispersed, cattle would have to adapt individually. This could also be tested by a nearest-neighbour test, which could be used to indicate if cattle remain as a herd. To support the failed ratio measurement, a confidence ratio could be calculated to further investigate if the cattle avoid or adapt to the virtual fence system [18].

To test if associative learning occurs through facilitation, a new experiment will have to be conducted. First, a period with experienced cattle where social facilitation can occur, followed by a trial period where the experienced cattle would be removed. This could be used to conclude if the cattle are experiencing social learning, that is, if the cattle can connect the warning and impulse when isolated.

The limitations of this study are the varying number of experienced cattle and the introduction method. These factors, which occur within the same case study, make it difficult to predict which factors are affecting the results, and therefore, it is impossible to reach any meaningful conclusion besides a difference within the case studies.

## 5. Conclusions

In conclusion, there was a significant difference in both the warnings and impulses per day between the learning- and the post-learning period for gradual introduction with no experienced cattle and instant learning with 1 experienced cow, but not for instant introduction with 10 experienced cattle. The case study with gradual introduction and no experienced cattle was the only case study that was able to increase the association between warning and impulse. Due to the non-standardised experiments, it was not possible to conclude if it was the experienced cattle, the introduction method or other parameters that caused the increased association between warning and impulse. Nevertheless, this study is the first to comprehensively investigate different learning methods and should encourage the design of future experiments to fully understand the effects of socialisation and introduction methods on cattle learning the virtual fence system.

## Figures and Tables

**Figure 1 animals-14-01456-f001:**
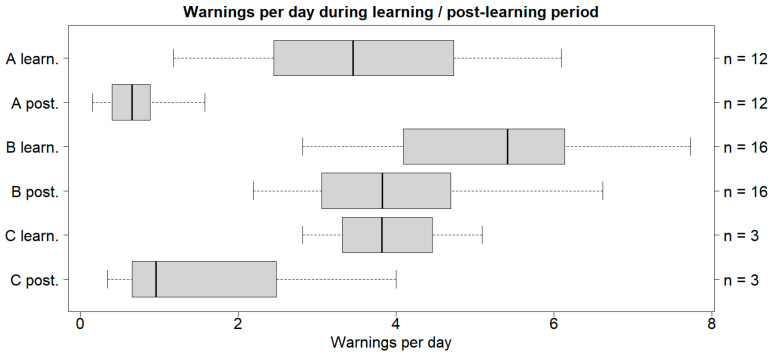
Box plot with the medians (25–75% quantiles) of the number of warnings per day given to the cattle. The *y*-axis shows the case study and period, while the *z*-axis shows the sample size. A refers to the cattle on Albuen, B refers to the cattle on Gåsehullerne, and C refers to the three cattle inserted on Albuen later. Note: *n* = sample size; learn. = learning period; post. = post-learning period.

**Figure 2 animals-14-01456-f002:**
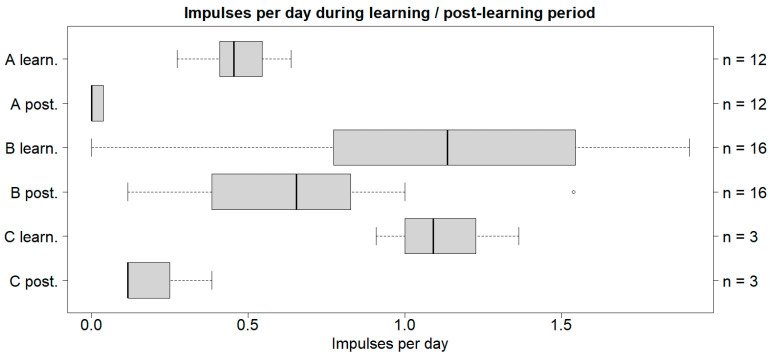
Box plot with the medians (25–75% quantiles) of the number of impulses per day given to the cattle. The *y*-axis shows the case study and period, while the *z*-axis shows the sample size. A refers to the cattle on Albuen, B refers to the cattle on Gåsehullerne, and C refers to the three cattle inserted on Albuen later. Note: *n* = sample size; learn. = learning period; post. = post-learning period.

**Figure 3 animals-14-01456-f003:**
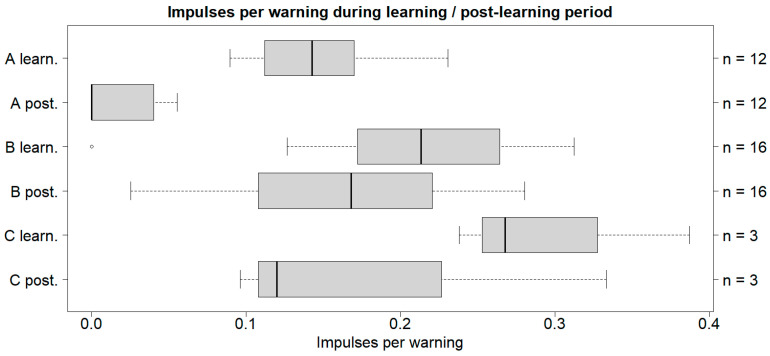
Boxplot with the medians (25–75% quantiles) of the impulses per warning given to the cattle. A refers to the cattle on Albuen, B refers to the cattle on Gåsehullerne, and C refers to the three cows inserted on Albuen later. Note: *n* = sample size; learn. = learning period; post. = post-learning period. A summary of each boxplot can be found in Appendix C.

**Table 1 animals-14-01456-t001:** Glossary.

Glossary	Definition
Experienced cow	A cow that has previously experienced the Nofence© system.
Learning period	The first 11 days of Nofence© introduction.
Post-learning period	The following 26 days after the end of the learning period.
Introduction method	The method the cattle are introduced to Nofence© with.
Warning	The sound emitted from the Nofence© collar when approaching the virtual border.
Impulse	The electrical impulse received from the Nofence© collar when crossing the virtual border.

**Table 2 animals-14-01456-t002:** Warnings per day during the learning- and post-learning period. A refers to the cattle on Albuen, B refers to the cattle on Gåsehullerne, and C refers to the three cattle inserted on Albuen later. Note: learn. = learning period; post. = post-learning period; MAD = median absolute deviation. The difference in the median was tested using the Wilcoxon rank-sum test. The difference in the variance was tested using Levene’s test. Significant values are indicated with an asterisk (*) when *p* < 0.05 for both tests.

	Comparison	Median ± MAD	Difference between Median (%)	Wilcoxon Rank-Sum Test	Levene’s Test
Warnings per day during learning- and post-learning period	A learn.	3.46 ± 1.62	136.7	4.64 × 10^−5^ *	0.0016 *
	A post.	0.65 ± 0.34			
	B learn.	5.41 ± 1.55	34.3	0.023 *	0.22
	B post.	3.83 ± 1.20			
	C learn.	3.82 ± 1.48	119.5	0.40	0.67
	C post.	0.96 ± 0.91			

**Table 3 animals-14-01456-t003:** Impulses per day during the learning period compared to impulses per warning during the post-learning period. A refers to the cattle on Albuen, B refers to the cattle on Gåsehullerne, and C refers to the three cattle inserted on Albuen later. Note: learn. = learning period; post. = post-learning period; MAD = median absolute deviation. The difference in the median was tested using the Wilcoxon rank-sum test. The difference in the variance was tested using Levene’s test. Significant values are indicated with an asterisk (*) when *p* < 0.05 for both tests.

	Comparison	Median ± MAD	Difference between Median (%)	Wilcoxon Rank-Sum Test	Levene’s Test
Impulses per day during learning- and post-learning period	A learn.	0.45 ± 0.14	-	2.28 × 10^−5^ *	0.0040 *
	A obs.	0			
	B learn.	1.14 ± 0.61	53.8	0.0053 *	0.14
	B obs.	0.65 ± 0.31			
	C learn.	1.09 ± 0.27	161.7	0.077	0.64
	C obs.	0.96 ± 0.91			

**Table 4 animals-14-01456-t004:** Impulses per warning during the learning period compared to impulses per warning during the post-learning period. A refers to the cattle on Albuen, B refers to the cattle on Gåsehullerne, and C refers to the three cattle inserted on Albuen later. Note: learn. = learning period; post. = post-learning period; MAD = median absolute deviation. The difference in the median was tested using the Wilcoxon rank-sum test. The difference in the variance was tested using Levene’s test. Significant values are indicated with asterisk (*) when *p* < 0.05 for both tests.

Comparison	Median ± MAD	Difference between Median (%)	Wilcoxon Rank-Sum Test	Levene’s Test
A learn.	0.14 ± 0.04	-	2.59 × 10^−5^ *	0.11
A post.	0			
B learn.	0.21 ± 0.07	21.1	0.076	0.76
B post.	0.17 ± 0.08			
C learn.	0.24 ± 0.04	66.6	0.40	0.72
C post.	0.12 ± 0.04			

## Data Availability

The data presented in this study are available on request from the corresponding author. The data are not publicly available because it will be part of future work.

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
