# Peer review of "Effects of Social Facilitation and Introduction Methods for Cattle on Virtual Fence Adaptation"

_animals, 2024, doi:10.3390/ani14101456_

Round 1

Reviewer 1 Report

Comments and Suggestions for Authors

Notes to the author:

The paper deals with a very interesting topic as it aims to prove the effectiveness of social facilitation and introduction methods for cattle to learn a virtual fencing system. Although in my opinion, and as mentioned by the authors, the experiments are too different to clearly identify an effect of either social facilitation or introduction method, the comparison of these two periods is very interesting and in gerneral, studies like this are missing until now. In particular, their experience with these different introduction methods and with experienced and inexperienced cattle is a valuable contribution to the field and should be helpful for further research in this area.

Abstract

LL 22-23 I do not understand the sentence. Which improvement do you mean? An improvement of virtual fences?

L 29 Please delete 'However' and rewrite the sentence. Please consider including the reason why it is necessary for livestock to be adapted to virtual fencing.

LL 36-37 Why do you call it learning and observation period? Were the animals observed during observation period?

LL 37 -38 What do you mean with 'impulses per warning'? To the best of my knowledge, the animal can't receive more than one impulse per warning. Are you dividing the total number of impulses by the total number of warnings? If so please write it.

Introduction

The introduction gives relevant but also some irrelevant information on virtual fencing. Please focus (after a short overview) on things that are relevant for the study.

LL 45-65 This section appears to be quite long. It is necessary to describe the background. But perhaps it would be appropriate to focus on the goals of the current study. Different types of training cattle to virtual fencing and social facilitation. Perhaps it would be possible to shorten the first part of the inroduction and to enlarge the second part of the introduction.

LL 88 ff. Please include ...were tested: i) Fewer impulses…...ii) Fewer impulses per warning…

LL 88-89 Please explain more precise what experienced cattle are. For example: ….(cattle which are grazed with Nofence the year before the trial)….

LL 90-91 Again please explain observation period and the way you calculate impulses per warning.

M & M

Detailed information on the experiment is provided by the authors. Although there are some issues:

Thank you for presenting a glossary. But what is the difference between Learning process and Learning? And why do you mention Impulse and not Warning?

L 97 Again why do you call it observation period?

LL 150 ff. This first period 6 May - … was an adaption period to grazing? If so please mention it.

LL 171 Statistical analysis or elsewhere: Please provide the formula of impulses per warning.

Please provide somewhere in M & M and/or elsewhere the reason why you choose ‚impulses per warning‘ and not a success ratio or something else. I assume you calculate it the same way Colusso et al. 2021 did. But please mention it.

This leads to another point. Although the methods you used for testing significant differences between medians are appropriate, it would be interesting to use a ‚glmmtmB‘ (R) to consider fixed and interaction effects. This could strenthgen your argumentation. I am not a statistican but I think using ‚glmmtmB‘ with the familiy ‚ordbeta‘ could be an approriate method to deal with the data and consider more effects. If you agree it is neccessary to use data between 0.00000….1 and 1 and therefore, I think you need another kind of ratio like success ration (Eftang et al.) ore confidence ratio (Hamidi et al. 2024).

Results

Perhaps it would improve the manuscript to start with an overview of the main results.

Caption table 1: Please add ‚Impulses per warning and day‘ or the appropriate abbreviaton for it in the table. You mention it in the caption but it needs to be mention in the table as well.

Figure 2: Please provide the explanation for lea. and obs.. I know what you mean but I think it is necessary to provide it due to a quick understanding.

Figure 5: The same.

Table 5 (caption): This is the kind of explanation for the abbreviations I mean ;) (lea. =learning…) Please add it to the places where it is missing.

Discussion

In general, the discussion has a good structure. Perhaps it would be easier to follow for the reader if you place the hypotheses at the top of the discussion. The results from the study are well connected with the current knowledge in the discussion. It also gives an outlook in how the knowledge generated in this study could be used.

However, there are some points to deal with:

Please discuss the approach to use impulses per warning. Why do you chose it, which are the advantages/disadvantages to succes or confidence ratio etc..

I acknowledge your critically discussion of your own results. Maybe it would be an improvement to make a new section called ‚limitations of our study‘ or something in this way.

L 296: I think ‚However‘ is not the right word for what you mean...If I understand it the right way, you agree with your first sentence.

LL 304-305: Good point

LL 308-309: Please reformulate the sentence. If you agree write: ‚This concludes that the total number of impulses and warnings could be reduced, apparently for reasons other than associative learning.‘

LL363-368: Please add that Hamidi et al. 2024 found differences between methods of measuring due to different kinds of learning. They found a) avoidance learning and b) sustainable learning could be measured by the use of different approaches to evaluate learning. Please discuss your approach to measure learning success in this context.

LL389-390 Very interesting idea!

LL 396-400 For me this sounds a bit like a conclusion.

Conclusions

Please do not only provide a summary but an outlook based on your results in the conclusion. The conclusion should stand for its own.

LL 408-411 These things are worth mentioning. But not in the conclusion. Please provide them in an own section of the discussion called ‚limitations of our study‘ or something in this way and discuss these points.

Comments on the Quality of English Language

The structure should be tightened up and the English improved (although I am not a native speaker)

Reviewer 2 Report

Comments and Suggestions for Authors

Simple summary: I’m wondering if ‘social facilitation’ is not a readily understood term in this context for a lay audience? Maybe instead it could state ‘…aims to assess social effects on learning between cattle with and without prior experience with virtual fencing’.

I’m also not clear on what you sentence at lines 22-24 means.

What is an improvement between a virtual fence and associative learning mean?

The simple summary is not very clear for being a lay person summary.

Line 30: adaptation to a virtual fence. Also on line 32: ‘a virtual fence’ or ‘virtual fencing’

Lines 33 and 34 and elsewhere in the abstract: write in the past tense as the study has been completed, not being proposed. There is currently a mix of past and present tense being used

Line 34: no need to label as ‘A, B, C’ if there is no other mention of the pastures in the abstract.

Line 35: it is really unclear on the experimental processes. I understand the abstract has a tight word limit, but I’m struggling to picture what was done. Was it 3 pastures, and 3 learning processes with one learning process per pasture? Then how was the variable of experienced/inexperienced included? Across all pastures/learning processes? Can you give numbers on the variation of ratios? How many animals were used in these trials? It is unclear if these were small or large groups you were working with. Was the learning process+pasture+inexperienced/experienced ratio all confounded together?

Line 36: I suggest to list ‘warnings’ first, since they come before the impulses

Line 36: what was the ‘learning period’ versus ‘observation period’? hours vs days? Days vs weeks?

If there is space within the abstract, it would be really helpful to give some numbers to ‘instant’ versus ‘gradual’ as it is not clear at the moment what this actually means. Did the introductions differ in hours, days, weeks…to be classified as ‘instant’ or ‘gradual’?

Line 45: major part of the world food industry….major contributor to European agriculture

Line 57: I think ‘temporary entanglement’ is a direct effect

Line 66: whose welfare does the VF increase? Wild animals? Needs to be stated clearer. ‘improving welfare’ would be more correct than ‘increasing welfare’.

Lines 71-72: I understand how the system works, but this statement is confusing. It is not explicit that the animal has to avoid the electric pulse by altering their behaviour in response to the warning tone.

Line 78: Studies also document

Line 85: improve animal welfare

I question the content of the introduction. While it is not incorrect, there are two paragraphs stating the benefits of virtual fencing, but the study is not about the benefits, it is about the learning process. The first mention of ‘welfare’ is in line 66, but it is presumed that is in reference to wild animal welfare based on the content of the first 2 paragraphs. If line 74 states many previous studies have found no substantial impact on animal welfare, then why is this study looking to improve animal welfare under virtual fencing technology? I can understand that fewer electric shocks is better for the animals, but where is the literature to lead into that aim? What about more detail on animal learning? The learning period? Length of time to learn? Evidence around social facilitation in previous studies? There is scope to make the introduction better suited to what this study is actually about, rather than a significant portion of the introduction justifying why VF is beneficial in agriculture systems (which is not what the study is about).

Line 91: I think ‘if the cattle had experienced a gradual introduction to’….would be more accurate wording now that I have read the methods. When you stated ‘raised with’, that implied they had experienced a virtual fence as calves through to adult cows.

Table 1: Please check the definitions for ‘Learning process’ and ‘Learning’ for clarity. Are both needed? What is the difference, besides the word ‘with’? What do you classify as ‘having learnt’ the virtual fence? The learning period is 11 days, why 11 days? How did you determine that timeframe? What is your criteria for ‘having learnt’?

Line 98: by ‘external operator’, do you mean someone from the research team (external to the farmer), or someone from Nofence (external to the research team)?

Line 104: Data were collected

Line 110: what do you mean by a ‘unitless’ number? Isn’t it a count? Can you not just state activity was measured as the number of steps?

Line 119: Three investigations at 2 locations doesn’t align with what is in the abstract. It states 3 pastures were used. Or is ‘pasture’ different from ‘location’?

Line 120: to the virtual fence

Line 122: the virtual fence

Line 124: was replaced with a virtual fence on all sides.

Line 127: or a small number of cattle from a field enclosed by an electrical fence were relocated to a pasture consisting of only virtual fences. (or only a virtual fence, depending on how many you had in place)

Line 137: The border of the virtual fence was moved 20 meters further south three times:

Line 152: why was the experienced cow excluded? What criteria was this based on? What did ‘review of the data entail’?

Line 155: On the same afternoon, the cattle were moved to a new 5.6 hectare area.

Line 157: In the following 26-day observation period

Lines 161 and 35: you state the methods were one gradual and two instant in the abstract, but the methods then describes 3 different processes where only one is described as ‘instant introduction’.

Line 172: The following tests were conducted:

Line 173: I suggest to always mention warnings first, as that is how the system operates. Warning before impulse.

Line 176: are you using ‘pasture’ here to refer to the different investigations? Or differences in pasture sizes when the pastures were expanded or cattle were moved to a new pasture in investigation B? It is not clear what the comparison is.

Line 177: no need to state ‘to investigate for significance’, that is redundant when you are applying a statistical test.

Line 178: same comment as above, does ‘pasture’ refer to ‘investigation’. The different terms in the methods vs abstract vs data section are confusing. Same comment also applies to the results. The ‘pasture’ is not really your main variable. There were differences in the introduction of the VF, the animals, and the pasture they were on.

Line 183: please present the results in the past tense.

Line 188: now they are being referred to as ‘herd’. Please be consistent in whether you identify the different assessments as ‘pasture’, ‘investigation’, ‘herd’ etc. The actual physical location of the trials, while stated in the methods, was not a key identifier, as you use ‘investigation’ rather than the location. So not sure if including place name in the legend is the best way to help the reader.

Lines 191-195: I disagree with the statistical comparisons between the three different investigations. There were so many variables that were different between them, so I’m not convinced you can conduct meaningful statistical comparisons about the learning periods of the different investigations. Other than to state they were different. But if you are making statistical comparisons to test the effect of the different learning processes, then this was not a controlled study that allows for that.

Table 1 at line 196: (should actually be Table 2). There is no definition of what A, B, and C refer to (comment applies to other tables where A, B, and C are not defined).

Line 210 onwards. I suggest presenting ‘warnings’ data first. Impulses come after warnings in the way the system operates, so the logical data presentation would be warnings first.

Line 219: this talks about impulses, but you just presented the results on warnings. The flow would be better with warnings presented first.

Line 224: impulses per warning is a pretty standard way of measuring learning across all VF studies to date, and is in your hypotheses at the end of the introduction. The text in the results reads like this conclusion was drawn as a result of your data presented to date.

Line 233: but didn’t you state earlier you removed the data of the experienced cow? How did that affect the results? Can you really state that was the ‘condition’, if that animal was excluded from the data?

Figure 3 legend. But there were so many other variables that differed. This legend implies the only thing that differed was the number of experienced cattle. But that was only one factor. And the factor that is highlighted depends on the data that you are presenting which is misleading to the reader.

Line 235: was significantly lower

Lines 237-239, and 260-261: This is interpretation, so better placed in the discussion rather than the results. And can this be concluded from this trial when there were so many variables that were different across your 3 groups? I think you can compare ‘learning vs observation’ for each group separately, and then make tentative comparisons across the 3 groups/investigations/pastures/herds in your discussion. But not statistical comparisons as this was not a controlled study to test your experimental variables of learning process and social facilitation.

Lines 279-280: see, here you now refer to the groups based on the learning process that was applied. But what about the experienced/inexperienced cattle? You cannot disentangle these factors at whim.

Section 4.1 The heading implies this is discussion on social facilitation. Yet a lot of the text is around the learning process (gradual/instant).

Lines 310-313: but there were so many other variables, so how can you properly conclude on the effect of experienced cattle inclusion or not? What about looking at patterns of where the animals actually were in the field? Were they next to each other to allow social facilitation to occur? If you are introducing inexperienced and unfamiliar cattle, what about the other social implications of this?

Line 325: collared not collard

Line 327: did you really predict that one experienced cow would have an impact? It would have to be a leader cow wouldn’t it to overcome the majority? Also, didn’t you state you removed the data from the experienced cow? So how does this affect your analyses/conclusions? Can you comment on where the experienced cow was in relation to the other animals in the group? Wouldn’t you need to look at spatial patterns of the herd to really understand why/why not for any effects of experienced cattle on learning rates?

Line 386: But you mention at lines 108-109 that the collars collected GPS and activity data. Why were these not included in the current study? You could look at GPS patterns if you have those data, eliminating the need to speculate on movement patterns in your study.

Lines 399-400. Not sure what you mean with this sentence. Are you stating the Nofence can apply to other livestock? Or apply to wildlife? How does a gradual introduction process that improves cattle welfare benefit wildlife management? The line of thought is not clear.

Line 404: We found a significant difference in

Lines 404-405: but it wasn’t just the learning process that differed – what about the inexperienced/experienced cattle factor?

Overall, I think the discussion will need to be significantly rewritten based on updated results/conclusions from the study.

Comments on the Quality of English Language

Suggestions have been made on where English language edits are needed, otherwise the manuscript is generally well written from a language point of view. 

Author Response

Please see the attachment, you comments are answered under "referee 2"

Round 2

Reviewer 1 Report

Comments and Suggestions for Authors

Thank you for the revised version which is significantly improved.

However, there are some minor comments:

LL 16 ff  M&M and limitations were mentioned in sufficient details. Please provide also the most important results of your study in the abstract.

L24 What does MC1 mean?

L25-26 Please provide a reference for this sentence.

L 30 ...installing, repairing and relocating...

LL 72-73 Due to my experience the collection of data were provided by the company on request to the researchers. Are you sure the collection of data from cattle collars were managed by the owner of the cattle?

LL 280-283 I do not undestand why you write that no significant reduction was found and in the next sentence 'it is not possible ....the reasons for this significant reduction ist.'  Do you mean: ....the reasons for the non-significant reduction.....?

LL 325 - 330 Consider to move this very interesting section in 4.3 or conclusion

Comments on the Quality of English Language

Although the quality of the English has improved, it would be an improvement to have it further edited by a native speaker.

Reviewer 2 Report

Comments and Suggestions for Authors

Simple summary: Virtual fencing is an alternative to the ubiquitously used physical fences. It uses GPS to determine the location of livestock relative to a virtual line.

Lines 40-41. You state here that previous studies have showed no substantial impact on animal welfare. So how does this then lead into the statement at line 54 about improving welfare. Where have you stated the gap and need for welfare improvement?

Line 104: with zero experienced cows)

Line 138: totalling 3 inexperienced cows.

Line 154-155: are you comparing between A, B, and C, or comparing the learning vs post-learning within A, B, and C separately. I disagree with comparing A, B, and C. They are too different for meaningful statistical comparison. What is the null hypothesis being tested? You have two overlapping factors, instant/gradual introduction and experienced/inexperienced. You cannot disentangle the two factors, therefore this experimental design does not allow you to test these hypotheses. This research is better presented as 3 case studies of virtual fencing application, highlighting the differences in fence introduction and experienced cattle, and pasture size etc. But it is not designed to test your hypotheses. Stating that the design did not allow us to actually test our hypotheses as a caveat at the end of the abstract, means you are stating the experiment is fundamentally flawed as your conclusion for not being able to confirm or deny your original hypotheses.

I’m wondering throughout if your abbreviation for ‘learning’ could be ‘learn’ rather than ‘lea’ given that lea is an actual word, and learn makes far more sense to the reader.

Lines 277-278: see, you automatically state your hypotheses cannot be confirmed or denied due to a fundamental flaw in your experimental design. This is not a ‘limitation’, this is an experimental design that prevented you from being able to test your specific hypotheses. Having these caveats in the discussion is not enough. The design is not suitable. The manuscript needs to be changed to account for this. Case studies on the application of VF, and the parameters that varied, then discussing those parameters and potential influence would still be worthy of publication. But having specific questions that were unable to be tested because your design was inadequate, is not the correct way to structure the paper.

I have not made further comments on the discussion as I think the manuscript still needs to be restructured before it is suitable for publication.

Comments on the Quality of English Language

Minor edits still required. 

Round 3

Reviewer 1 Report

Comments and Suggestions for Authors

Thanks for this carefully revised version which should be acceptable for publication after two minor improvements.

LL 291-292 The citation of Hamidi et al. 2024 (18) appeared to be too late. Please consider moving it from L 292 to L 291 after ".....sustainable learning [18]. " as Hamidi et al. did not know anything about your studydesign.

Conclusion LL 336-339 I do not fully understand the sentence. However, it is a long sentence about things you have already said in other parts of the manuscript. Please consider to say something that acknowledged your work and gives an outlook.I don not think that the current sentence provide this.

If you agree, you could perhaps write something like: "Nevertheless, this study is the first to so comprehensively compare different learning methods and should encourage the design of future experiments to fully understand the effects of socialisation and introduction methods on cattle learning the virtual fence system."

Comments on the Quality of English Language

Although the English language is improved language editing by a native speaker would be a further improvement.

Reviewer 2 Report

Comments and Suggestions for Authors

The revisions have improved the manuscript so the findings are not over-stated. 

Author Response

We thank you for your help reviewing this paper.

Sincerely,
The authors